# On the Usefulness of Embeddings, Clusters and Strings for Text Generator Evaluation

**Tiago Pimentel**[*,1], **Clara Meister**[*,2], **Ryan Cotterell**[2]
[1] University of Cambridge, [2] ETH Zürich
tp472@cam.ac.uk, {clara.meister, ryan.cotterell}@inf.ethz.ch

## Abstract

A good automatic evaluation metric for language generation ideally correlates highly with human judgements of text quality. Yet, there is a dearth of such metrics, which inhibits the rapid and efficient progress of language generators. One exception is the recently proposed MAUVE. In theory, MAUVE measures an information-theoretic divergence between two probability distributions over strings: one representing the language generator under evaluation and the other representing the true natural language distribution. MAUVE's authors argue that its success comes from the qualitative properties of their proposed divergence. Yet in practice, as this divergence is uncomputable, MAUVE approximates it by measuring the divergence between multinomial distributions over clusters instead, where cluster assignments are attained by grouping strings based on a pre-trained language model's embeddings. As we show, however, this is not a tight approximation—in either theory or practice. This begs the question: why does MAUVE work so well? In this work, we show that MAUVE was right for the wrong reasons, and that its newly proposed divergence is not necessary for its high performance. In fact, classical divergences paired with its proposed cluster-based approximation may actually serve as better evaluation metrics. We finish the paper with a probing analysis; this analysis leads us to conclude that—by encoding syntactic- and coherence-level features of text, while ignoring surface-level features—such cluster-based substitutes to string distributions may simply be better for evaluating state-of-the-art language generators.[1]

## 1 Introduction

Probabilistic text generators have improved greatly over the last years, with models producing increasingly human-like text (Yang et al., 2019; Brown et al., 2020; Raffel et al., 2020; Rae et al., 2021; Hoffmann et al., 2022). As the gap between human and model-generated text closes, the quality of our evaluation metrics becomes ever more important for determining generator quality, especially given the increasing number of user-facing systems employing these generators. While human evaluations serve as the gold standard, they are costly (in both time and money), leading researchers to rely on automatic metrics—i.e., metrics that can be measured by a computer—for the bulk of their development process.

Many automatic language generator evaluation metrics share the same underlying mechanism: the quantitative comparison of two probability distributions. Specifically, most metrics measure a difference between the distributions over strings defined by: (1) a language generation model[2] and (2) the natural language itself. This includes some of the most widely used language evaluation metrics:[3] cross-entropy (Shannon, 1948), perplexity (Jelinek et al., 1977), and (more recently) MAUVE (Pillutla et al., 2021). As typically applied to evaluate language generators, however, these metrics have a number of computational and qualitative issues (discussed in §3). Such issues manifest empirically: the most commonly used automatic metrics are known to correlate poorly with human judgements (Wiseman et al., 2017; Reiter, 2018; Sellam et al., 2020; Gehrmann et al., 2021).

---

[*]Equal contribution.

[1]Code available at `https://github.com/rycolab/clusters-in-language-evaluation`.

[2]We define a language generator as a probability distribution $q_\mathbf{w}$ over strings $\mathbf{w}$. Specifically, we consider this distribution as used during generation. E.g., if decoding is performed with nucleus sampling, we consider the final distribution where every sentence with tokens not in the nucleus is assigned a probability of 0.

[3] Most measures we consider are not metrics in a strict sense; we use the term "metric" out of convention.

A newly proposed metric stands apart: MAUVE (Pillutla et al., 2021). In theory, MAUVE measures the area under the curve formed by the divergence between two probability distributions, qualitatively mimicking a precision–recall quantification (Djolonga et al., 2020; Kynkäänniemi et al., 2019). The authors attribute the success of their metric to the qualitative properties of this new class of divergences. Yet, due to this divergence being in practice uncomputable, Pillutla et al. propose an approximation to it. Specifically, rather than directly comparing the original two distributions over strings, MAUVE first clusters samples taken from these distributions based on the embeddings of a pre-trained language model; it then estimates the proposed divergence using the samples' empirically-observed multinomial distributions over cluster assignments. As we will show, this approximation is bad in both theory §4 and practice §5.1—to the point that the term "approximation" is arguably a misnomer. Thus, the reasons why MAUVE works well—knowledge which is important for the continued progress of language generator evaluation metrics—are still unknown.

In this work, we aim to uncover these reasons. To this end, we consider the axes on which MAUVE differs from other evaluation metrics: is MAUVE's success due to the new divergence metric, to its "approximation", or both? Empirically, we identify MAUVE's substitution of probability distributions over strings with probability distributions over embedding-based clusters as the main factor for its success. We show that mathematically, this substitution leads to a quite biased estimator of the original string-based divergences. Yet it also leads to metrics with lower variance and stronger correlations with human judgements. In fact, all divergence measures analysed here correlate more strongly with human judgements when cluster-based distributions are used in place of string-based ones.

Finally, in order to understand the root of the effectiveness of these cluster-based metrics, we probe the clusters themselves. We find that sentence-level permutations within texts noticeably affect cluster assignments, suggesting that cluster-based metrics are susceptible to attributes such as coherence. On the other hand, basic manipulations that render text unhuman-like, such as removing all articles from the input text, do not seem to affect these metrics significantly. Together, these results lead us to conjecture that embedding-based metrics may be favourable when estimating the quality of state-of-the-art (SOTA) language generators, as SOTA models are known to (at least typically) produce grammatical text. That is, by ignoring surface-level features of text—while emphasising discourse- and coherence-level ones—clustered embeddings may simply be better suited for the evaluation of the top language generation systems. Yet these findings also suggest routes through which such metrics can be gamed, bringing into question their robustness. We believe these findings, along with the theoretical framework we provide for evaluation metrics' comparison, are important for the further development of language generator evaluation metrics.

## 2 DIVERGENCE METRICS FOR LANGUAGE GENERATOR EVALUATION

When evaluating language generation systems, we will first assume the existence of an unknown ground-truth distribution $p_{\mathbf{w}}$. This distribution is defined over strings $\mathbf{w}$ and its domain spans $\mathcal{W} \equiv \Sigma^*$, where $\Sigma$ is an alphabet of words and $\Sigma^*$ is its Kleene closure. Second, we are given a probabilistic text generator $q_{\mathbf{w}}$, which is also a distribution over $\mathcal{W}$. An evaluation metric for a language generator $q_{\mathbf{w}}$ can now be defined as a measure of its "distance" from $p_{\mathbf{w}}$: $\Delta(p_{\mathbf{w}}, q_{\mathbf{w}})$. In short, $\Delta(\cdot, \cdot)$ should return high values if $q_{\mathbf{w}}$ is a bad approximation to $p_{\mathbf{w}}$, and it should return low values if it is a good one.

Notably, it is not clear whether $q_{\mathbf{w}}$ being a good approximation to $p_{\mathbf{w}}$ in terms of an arbitrary $\Delta(\cdot, \cdot)$ guarantees that it will be a good language generator. Indeed, models that perform well in terms of standard metrics, such as perplexity, often still produce poor-quality text (Holtzman et al., 2020). Thus, we are interested specifically in $\Delta(\cdot, \cdot)$ that correlate highly with human quality judgements.

More formally, we define human quality judgements as a (potentially noisy) mapping $\boldsymbol{\alpha}(q_{\mathbf{w}})$ from a language generator to a real-valued score. For fixed $p_{\mathbf{w}}$, a useful metric $\Delta(p_{\mathbf{w}}, \cdot)$ for evaluating the quality of a language generator $q_{\mathbf{w}}$ is one whose scores correlate highly with $humanscores(\cdot)$. This notion can be operationalised as follows. Assume we have $N$ language generator models. Let us define:

$$\boldsymbol{\delta}_{\texttt{human}}(q_{\mathbf{w}}^{(1)}, \ldots, q_{\mathbf{w}}^{(N)}) = \left[\boldsymbol{\alpha}\left(q_{\mathbf{w}}^{(1)}\right), \ldots, \boldsymbol{\alpha}\left(q_{\mathbf{w}}^{(N)}\right)\right] \tag{1}$$

$$\boldsymbol{\delta}_{\texttt{metric}}(q_{\mathbf{w}}^{(1)}, \ldots, q_{\mathbf{w}}^{(N)}) = \left[\Delta\left(p_{\mathbf{w}}, q_{\mathbf{w}}^{(1)}\right), \ldots, \Delta\left(p_{\mathbf{w}}, q_{\mathbf{w}}^{(N)}\right)\right] \tag{2}$$

We then quantify a metric's usefulness on a specific natural language task (and its distribution $p_{\mathbf{w}}$) as:

$$\texttt{quality}(\Delta, p_{\mathbf{w}}) = |\texttt{corr}\left(\boldsymbol{\delta}_{\texttt{human}}, \boldsymbol{\delta}_{\texttt{metric}}\right)| \tag{3}$$

We now review common choices for $\Delta(\cdot, \cdot)$. Given the probabilistic nature of most language generators, a number of divergence measures are among these choices, which quantify the difference between two probability distributions.[4] The rest of this work focuses primarily on this class of metrics.

**Forward Divergence.** Cross-entropy, $\Delta_{\mathrm{H}}(p_{\mathbf{w}}, q_{\mathbf{w}}) \overset{\text{def}}{=} \mathrm{H}(p_{\mathbf{w}}, q_{\mathbf{w}})$, which is equivalent (up to an additive constant) to the forward Kullback–Leibler (KL) divergence, is one such choice:

$$\Delta_{\rightarrow}(p_{\mathbf{w}}, q_{\mathbf{w}}) \overset{\text{def}}{=} \mathrm{KL}(p_{\mathbf{w}} \mid\mid q_{\mathbf{w}}) = \mathrm{H}(p_{\mathbf{w}}, q_{\mathbf{w}}) - \mathrm{H}(p_{\mathbf{w}}) \overset{(1)}{\lhd} \mathrm{H}(p_{\mathbf{w}}, q_{\mathbf{w}}) = \Delta_{\mathrm{H}}(p_{\mathbf{w}}, q_{\mathbf{w}}) \tag{4}$$

where we use $\lhd$ to signify additive or multiplicative equivalence. (1) is true since $\mathrm{H}(p_{\mathbf{w}})$ is constant with respect to $q_{\mathbf{w}}$. Since Pearson and Spearman correlations—the metrics we use to evaluate $\Delta$'s quality—are invariant to translational shifts, the cross-entropy and forward KL are equivalent as language generator metrics. We will refer to them interchangeably during subsequent comparisons.

**Backward Divergence.** Albeit much less common, another potential evaluation metric would be the backward (exclusive) KL divergence:

$$\Delta_{\leftarrow}(p_{\mathbf{w}}, q_{\mathbf{w}}) \overset{\text{def}}{=} \mathrm{KL}(q_{\mathbf{w}} \mid\mid p_{\mathbf{w}}) \tag{5}$$

As opposed to the forward KL, when use as an evaluation metric, Eq. (5) is not effectively equivalent to the cross-entropy between $q_{\mathbf{w}}$ and $p_{\mathbf{w}}$, as $\mathrm{H}(q_{\mathbf{w}})$ is not constant across language generators $q_{\mathbf{w}}$.

**Exponentiated Divergence.** By far, the most common choice of $\Delta$ to evaluate language models is the perplexity: $\Delta_{\mathrm{perp}}(p_{\mathbf{w}}, q_{\mathbf{w}}) \overset{\text{def}}{=} e^{\mathrm{H}(p_{\mathbf{w}}, q_{\mathbf{w}})}$. Notably, perplexity is equivalent (up to a multiplicative constant) to an exponentiated Kullback–Leibler divergence between $p_{\mathbf{w}}$ and $q_{\mathbf{w}}$, which follows from the same relationship as in Eq. (4). Given the property that both Pearson and Spearman correlations are invariant to a change in scale, the perplexity and exponentiated KL will thus be equivalent as language generator metrics. For consistency, we will use solely the exponentiated KL in our analyses:

$$\Delta_{\mathrm{exp}}(p_{\mathbf{w}}, q_{\mathbf{w}}) \overset{\text{def}}{=} e^{\mathrm{KL}(p_{\mathbf{w}}||q_{\mathbf{w}})} \tag{6}$$

**Jensen–Shannon Divergence.** Note that the KL divergence is non-symmetric and unbounded. On the other hand, the Jensen–Shannon (JS) divergence—defined as the average of two KLs—is symmetric with respect to its inputs and is guaranteed to produce bounded values:

$$\Delta_{\mathrm{JS}}(p_{\mathbf{w}}, q_{\mathbf{w}}) \overset{\text{def}}{=} \frac{1}{2}\left(\mathrm{KL}(p_{\mathbf{w}} \mid\mid r_{\mathbf{w}}^{.5}) + \mathrm{KL}(q_{\mathbf{w}} \mid\mid r_{\mathbf{w}}^{.5})\right), \qquad r_{\mathbf{w}}^{\lambda} = \lambda\, p_{\mathbf{w}} + (1 - \lambda)\, q_{\mathbf{w}} \tag{7}$$

**Area Under the Curve (AUC) Divergence.** Finally, information divergence frontiers are a recently proposed class of metrics for generative models (Sajjadi et al., 2018; Kynkäänniemi et al., 2019). The variant proposed by Pillutla et al. (2021) computes the area under the curve formed by a series of Kullback-Leibler divergences as we change a mixing parameter $\lambda$:

$$\Delta_{\mathrm{AUC}}(p_{\mathbf{w}}, q_{\mathbf{w}}) = 1 - \mathrm{AUC}\left(e^{-s\,\mathrm{KL}(p_{\mathbf{w}}||r_{\mathbf{w}}^{\lambda})}, \; e^{-s\,\mathrm{KL}(q_{\mathbf{w}}||r_{\mathbf{w}}^{\lambda})}\right), \qquad r_{\mathbf{w}}^{\lambda} = \lambda\, p_{\mathbf{w}} + (1 - \lambda)\, q_{\mathbf{w}} \tag{8}$$

where $\lambda$ is varied across the interval $[0, 1]$, and $s \in \mathbb{R}_{>0}$ is a strictly positive real-valued scaling constant. Note that we define the AUC divergence as $1 - \mathrm{AUC}(\cdot, \cdot)$ so that a larger value indicates a greater discrepancy with the reference corpus $p_{\mathbf{w}}$.

## 3 INFELICITIES AND APPROXIMATIONS

There are several issues, both computational and qualitative, with using the divergences presented in §2 to evaluate language generators. We now review these issues, along with both commonly-used and newly-proposed methods to address them via approximations.

---

[4]Here we make use of *shifted* divergences: a divergence measure that potentially has an additive constant, i.e., there exists a constant $c \in \mathbb{R}$ such that $\Delta(\cdot, \cdot) + c$ is a divergence. For our purposes, an additive constant should not affect the quality of our metrics, as the correlation in Eq. (3) is translation-invariant.

### 3.1 Necessity of Full Support

A well-known property of the (forward) KL divergence between two distributions $p_{\mathbf{w}}$ and $q_{\mathbf{w}}$ is that it is infinite for any $q_{\mathbf{w}}$ that assigns 0 probability to an event in the support of $p_{\mathbf{w}}$ (i.e., for which $p_{\mathbf{w}}(\mathbf{w}) > 0$). The above is often not an issue for $\Delta_{\exp}$ and $\Delta_{\rightarrow}$: most neural language generators cannot assign 0 probability to *any* string due to the final softmax operation typically used to project their outputs onto the probability simplex. However, these same models are often used with decoding strategies that prune the space $\mathcal{W}$: e.g., both top-$k$ and nucleus sampling modify $q_{\mathbf{w}}$ such that strings which do not meet a certain criterion are reassigned 0 probability. While top-$k$ and nucleus sampling typically lead to systems with qualitatively better text, they will likely be given an infinitely bad score by both $\Delta_{\exp}$ and $\Delta_{\rightarrow}$, which is perhaps too harsh a penalty for an otherwise good language generator.

### 3.2 $p_{\mathbf{w}}$ is Unknown

In practice, we do not have access to the true distribution $p_{\mathbf{w}}$. Rather, we are typically given a corpus $\{\mathbf{w}_n^{p_{\mathbf{w}}}\}_{n=1}^{N}$, whose instances we assume to be sampled i.i.d. from $p_{\mathbf{w}}$. The common approach to address this issue is thus to derive a statistical estimator $\widehat{\Delta}$ that uses this corpus to approximate $\Delta$. There are two common strategies for building such estimators: Monte Carlo and plug-in estimation.

**Monte Carlo Estimation.** Our i.i.d. assumption w.r.t. samples in $\{\mathbf{w}_n^{p_{\mathbf{w}}}\}_{n=1}^{N}$ allows us to derive a Monte Carlo estimator for certain divergences. We start with the forward KL divergence:

$$\widehat{\mathrm{KL}}(p_{\mathbf{w}} \| q_{\mathbf{w}}) \stackrel{\text{def}}{=} \frac{1}{N} \sum_{n=1}^{N} \log \frac{p_{\mathbf{w}}(\mathbf{w}_n^{p_{\mathbf{w}}})}{q_{\mathbf{w}}(\mathbf{w}_n^{p_{\mathbf{w}}})} = -\frac{1}{N} \sum_{n=1}^{N} \log q_{\mathbf{w}}(\mathbf{w}_n^{p_{\mathbf{w}}}) + \texttt{const} \tag{9}$$

where $\texttt{const} \in \mathbb{R}$ is constant with respect to $q_{\mathbf{w}}$. Eq. (9) is an unbiased estimator of KL divergence, which in turn allows us to build estimators $\widehat{\Delta}_{\rightarrow}$ and $\widehat{\Delta}_{\exp}$. Unfortunately, unbiased estimates of $\Delta_{\leftarrow}$, $\Delta_{\mathrm{JS}}$ and $\Delta_{\mathrm{AUC}}$ are not as straightforward to compute, as they require explicit knowledge of $p_{\mathbf{w}}$ rather than just samples (see App. A). This issue motivates the use of our next set of estimation techniques.

**Plug-in Estimation.** Here we consider estimation via building an approximation of $p_{\mathbf{w}}$ itself to use in the formulas given in §2. Specifically, we construct a density estimator for $p_{\mathbf{w}}$ (which we denote as $\widehat{p}_{\mathbf{w}}$) and "plug it into" a given $\Delta$.[5] However, this is a bit circular: the task of building a language generator $q_{\mathbf{w}}$ itself is often framed as density estimation of $p_{\mathbf{w}}$. Thus, if we think $q_{\mathbf{w}}$ is the "best" estimator for $p_{\mathbf{w}}$, we should logically use it in our plug-in estimator. Yet, using $q_{\mathbf{w}}$ would be nonsensical; by the definition of a (shifted) divergence, it would always lead to the lowest possible value of $\Delta$, e.g., $\Delta_{\rightarrow}(q_{\mathbf{w}}, q_{\mathbf{w}}) = 0$. To use plug-in estimation in this setting, we should therefore choose a different estimator for $p_{\mathbf{w}}$, e.g., from a family of density estimators that differs from those used to create $q_{\mathbf{w}}$. More formally, we consider a function $\pi$ which takes a corpus as input and produces a (queryable) distribution $\widehat{p}_{\mathbf{w}} \stackrel{\text{def}}{=} \pi(\{\mathbf{w}_n^{p_{\mathbf{w}}}\}_{n=1}^{N})$. This function typically induces a secondary model, e.g., an $n$-gram model or neural network, trained on the corpus $\{\mathbf{w}_n^{p_{\mathbf{w}}}\}_{n=1}^{N}$.

Our chosen $\pi$ may introduce biases (e.g., from the inductive biases of the architecture parameterising $\pi$) into our metrics' estimation. To balance out such biases, we may consider using the same method to create an approximation $\widehat{q}_{\mathbf{w}}$ for use in our plug-in estimators, rather than directly querying $q_{\mathbf{w}}$:

$$\widehat{\Delta}_{\leftarrow}(\{\mathbf{w}_n^{p_{\mathbf{w}}}\}_{n=1}^{N}, q_{\mathbf{w}}) \stackrel{\text{def}}{=} \widehat{\mathrm{KL}}(\widehat{q}_{\mathbf{w}} \| \widehat{p}_{\mathbf{w}}) \tag{10}$$

Plug-in estimators for $\Delta_{\mathrm{JS}}$ and $\Delta_{\mathrm{AUC}}$ are defined similarly. Further, if $\widehat{q}_{\mathbf{w}}$ is a smoothed approximation to the original $q_{\mathbf{w}}$, using it may also mitigate the issues discussed in §3.1. We thus also compute estimators for the forward/exponentiated divergences using plug-in estimators, e.g.,:

$$\widehat{\Delta}_{\rightarrow}(\{\mathbf{w}_n^{p_{\mathbf{w}}}\}_{n=1}^{N}, q_{\mathbf{w}}) \stackrel{\text{def}}{=} -\frac{1}{N} \sum_{n=1}^{N} \log \widehat{q}_{\mathbf{w}}(\mathbf{w}_n^{p_{\mathbf{w}}}) \tag{11}$$

---

[5]Often, plug-in and Monte Carlo estimators must be used together. Even if we are measuring the divergence between two queryable distributions, the sum over $\mathcal{W}$ is infinite and non-decomposable, thus uncomputable.

Unfortunately, most functions $\pi$ cannot produce a good estimate of $p_{\mathbf{w}}$ using only a small corpus, which is the case we consider since we rely on evaluation sets for $\{\mathbf{w}_n^{p_{\mathbf{w}}}\}_{n=1}^{N}$. While the best available language models are a class of $\pi$ typically trained on millions (if not billions) of sentences, a standard evaluation set is quite small—on the order of one to ten thousand sentences—and we cannot expect $\pi$ to provide a good $\widehat{p}_{\mathbf{w}}$ when fit using only such a small dataset. Accordingly, depending on our choice of $\pi$, this class of metrics may be either high variance or high bias, both of which are problematic.

### 3.3 CLUSTERING-BASED APPROXIMATIONS

For the $\widehat{\Delta}$ above that require density estimators for $p_{\mathbf{w}}$ and/or $q_{\mathbf{w}}$, our choice of $\pi$ will have a large effect on its value. We may thus wish to rethink our approximation technique altogether, and instead work with different distributions for which we can create lower variance density estimators. This is the approach used by Pillutla et al. (2021) when approximating $\Delta_{\text{AUC}}$. Specifically, instead of computing the above metrics on the original distributions $p_{\mathbf{w}}$ and $q_{\mathbf{w}}$, they use the cluster-based distributions $p_c$ and $q_c$. Given a pre-trained language model, these cluster-based distributions are defined as:

$$p_c(c) = \sum_{\mathbf{w} \in \mathcal{W}} p_{\mathbf{w}}(\mathbf{w}) \, \mathbb{1} \left\{ c = \phi(\mathsf{PLM}(\mathbf{w})) \right\} \tag{12}$$

where $\mathsf{PLM}(\cdot)$ takes as input an utterance $\mathbf{w}$ and outputs an embedding $\mathbf{r}$, and $\phi(\cdot)$ is a pretrained clustering function. Note that the function $\phi(\cdot)$ is trained jointly on samples from the two distributions under consideration as we will detail later; we defer the reader to Pillutla et al. (2021) for a more detailed explanation of the procedure. Given these distributions, we can evaluate cluster-based versions of all the divergences above, simply by substituting the original $p_{\mathbf{w}}$ and $q_{\mathbf{w}}$ with the new $p_c$ and $q_c$.

## 4 ANALYSING CLUSTER-BASED APPROXIMATIONS

We now take a closer look at the biases introduced by the substitution of cluster-based distributions suggested in §3.3. For simplicity, we focus on the bias introduced to $\text{KL}(p_{\mathbf{w}} \,||\, q_{\mathbf{w}})$—a computation involved in MAUVE's $\Delta_{\text{AUC}}$. This divergence can be decomposed as:

$$\underbrace{\text{KL}(p_{\mathbf{w}} \,||\, q_{\mathbf{w}})}_{\texttt{string-based KL}} \stackrel{(1)}{=} \text{KL}(p(c) \,||\, q(c)) + \underbrace{\text{KL}(p(\mathbf{w} \mid c) \,||\, q(\mathbf{w} \mid c))}_{\geq 0} \geq \underbrace{\text{KL}(p_c \,||\, q_c)}_{\texttt{cluster-based KL}} \tag{13}$$

where (1) follows from the fact that $p(c, \mathbf{w}) = p(\mathbf{w})$, which is true because the cluster assignment is deterministic, i.e.: $p(c \mid \mathbf{w}) = \mathbb{1}\{c = \phi(\mathsf{PLM}(\mathbf{w}))\}$. See the full decomposition of this equation in App. B. Notably, as KL divergences are always non-negative, the cluster-based version is negatively biased, lower-bounding the string-based one. Further, the actual measurement is done on the distribution over cluster assignments $p(c)$; the distribution $p(\mathbf{w} \mid c)$ is completely ignored.

Assuming a reasonable number of clusters is used when defining $p_c$, however, it should be easier to create good approximations of distributions over clusters than distributions over strings due to the sheer difference in the size of the supports alone. Consequently, the variance of cluster-based metrics should be lower, at the cost of the bias introduced by this substitution. Further, it is not clear whether this bias is inherently bad when evaluating the quality of language generators: the answer to this question must be determined empirically (by measuring the correlation in Eq. (3)). To this end, we now provide an empirical comparison between string- and cluster-based language generation evaluation.

## 5 EXPERIMENTS

**Setup.** We follow the setup of Pillutla et al. throughout our experiments. We compare systems for open-ended text generation $q_{\mathbf{w}}$ with human-generated text $p_{\mathbf{w}}$. As human-generated samples, we use $5k$ strings taken from WebText's test set. As model-generated text, we sample $5k$ strings from each of our evaluated systems, conditioning our models on the first 10 words of human-generated strings before sampling (i.e., a text-completion task). For our language generators, we compare 4 model architectures (all variants of GPT-2), each under two decoding strategies, giving us a total of 8 systems. Explicitly, we compare the `small`, `medium`, `large`, and `XL` versions of GPT-2, decoding strings using either ancestral or nucleus sampling. Following Pillutla et al. (2021), we use a nucleus

probability of $0.9$ for `small` and `medium`, while $0.95$ for `large` and `XL` GPT-2's. Importantly, we run our experiments only on English text, which is a notable limitation of our work; future work should verify that findings hold across languages.

**String-based Approximations $\widehat{p}_{\mathbf{w}}$.** To compute our string-based divergences, we require a secondary language model $\widehat{p}_{\mathbf{w}}$ to estimate $p_{\mathbf{w}}$. Further, following the issues highlighted in §3, we will also rely on a secondary language model $\widehat{q}_{\mathbf{w}}$ to estimate $q_{\mathbf{w}}$. We will use $n$-gram models for these approximations. Specifically, we use Kneser-Essen-Ney smoothed 5-gram models, as implemented in KenLM (Ney et al., 1994; Heafield, 2011). We choose $n$-gram models explicitly because—while they are by no means SOTA language models—they should have inductive biases which are different from the models we are trying to evaluate. We present results using LSTM-based estimators in App. E. When computing $\Delta_{\mathrm{AUC}}$, we use a scaling constant $s$ of $0.2$.

**Cluster-based Approximations $\widehat{p}_c$.** Cluster-based distributions, as presented in Eq. (12), are defined by a choice of PLM($\cdot$) and pre-trained clustering function $\phi(\cdot)$. We rely on GPT-2 `XL` as our PLM, and use $K$-means as our clustering function; results using other pre-trained language models are in App. E. Specifically, we first extract embeddings from the last word in each sentence using GPT-2 `XL` and then use PCA to reduce their dimensionality (keeping 90% of the original variance explained); results using mean of word embeddings can be found in App. E. We then train $K$-means (with $K = 500$) on a joint set of GPT-2 embeddings extracted from: the $5k$ human-generated strings, and $5k$ model-generated sentences. Finally, we approximate $\widehat{p}_c$ and $\widehat{q}_c$ by computing the frequency with which strings (among these $5k$ used ones) are assigned to each cluster. To avoid infinite divergence measures, we estimate distributions using Laplace smoothing with $\alpha = 1$ (which is equivalent to imposing a Dirichlet distributed prior with $\alpha = 1$ over the cluster allocation). When computing $\Delta_{\mathrm{AUC}}$, we use a scaling constant $s$ of $5$.[6][7]

## 5.1 Does $p_c$ Approximate $p_{\mathbf{w}}$?

Our first experiment tries to identify whether $p_c$ and $q_c$ provide faithful approximations of $p_{\mathbf{w}}$ and $q_{\mathbf{w}}$. To this end, we compare both $\widehat{q}_{\mathbf{w}}$ and $\widehat{q}_c$ to the true $q_{\mathbf{w}}$, i.e., the language generator under evaluation. Explicitly, we compute the Spearman correlations between the probabilities assigned by each model to the strings in $\{\mathbf{w}_n^{q_{\mathbf{w}}}\}_{n=1}^N$.

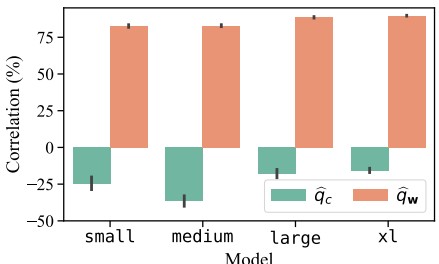

Figure 1: Correlations between the true $q_{\mathbf{w}}$ and the estimated $\widehat{q}_c$ and $\widehat{q}_{\mathbf{w}}$.

Fig. 1 presents these correlations. We see that—despite being estimated on very little data—probability estimates from our $n$-gram models correlate strongly with the ground-truth probabilities of $q_{\mathbf{w}}$; this result holds for all four GPT-2 architectures. On the other hand, our cluster-based probabilities consistently present *negative* correlations with $q_{\mathbf{w}}$. This result has an important implication: if cluster distributions do not correlate with $q_{\mathbf{w}}$, then $\mathrm{KL}(\widehat{p}_c \mid\mid \widehat{q}_c)$ is likely a poor estimate of $\mathrm{KL}(p_{\mathbf{w}} \mid\mid q_{\mathbf{w}})$. This further implies that the approximation used by Pillutla et al. is not an accurate estimate of $\Delta_{\mathrm{AUC}}(p_{\mathbf{w}}, q_{\mathbf{w}})$, which brings into question whether this new divergence is really responsible for MAUVE's success.

## 5.2 $\Delta$ as Text Evaluation Metrics

We now compare how various string- and cluster-based divergence measures correlate with human judgement scores.[8] In short, Fig. 2 shows that all divergences do better when estimated with cluster distributions. These results evince that MAUVE's (Pillutla et al., 2021) high correlations with human judgements (represented here as $\widehat{\Delta}_{\mathrm{AUC}}(p_c, q_c)$) are mainly due to their use of cluster-based approximations ($p_c, q_c$), rather than to their proposed divergence $\Delta_{\mathrm{AUC}}$. In fact, we see

---

[6]We ran our entire pipeline (from sampling strings $\{\mathbf{w}_n^{q_{\mathbf{w}}}\}_{n=1}^N$ from $q_{\mathbf{w}}$ to evaluating the KLs) with 5 different seeds. The variance across seeds is depicted as error bars in Figs. 1 and 2.

[7]We follow Pillutla et al.'s (2021) experimental setup here. Note that Pillutla et al. provide an in-depth analysis of experimental design choices and show MAUVE is robust to various hyperparameter choices.

[8]We use the human judgement scores collected by Pillutla et al. (2021).

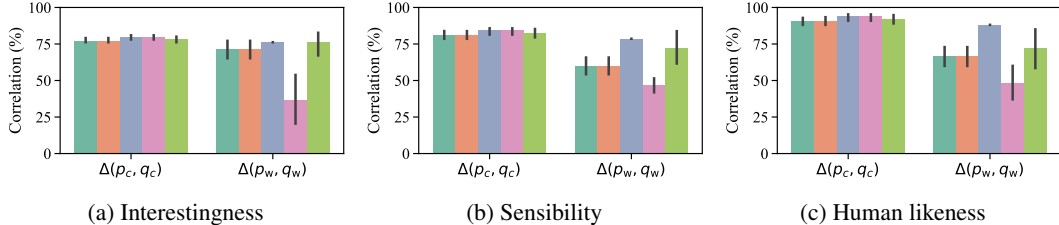

Figure 2: Correlations between string- and cluster-based divergences with human judgement scores. Legend: $\Delta_{\mathrm{exp}}$ **in dark green**; $\Delta_{\rightarrow}$ **in orange**; $\Delta_{\leftarrow}$ **in blue**; $\Delta_{\mathrm{JS}}$ **in pink**; $\Delta_{\mathrm{AUC}}$ **in lime green**.

slight improvements over $\widehat{\Delta}_{\mathrm{AUC}}$ when using the divergences $\widehat{\Delta}_{\leftarrow}$ and $\widehat{\Delta}_{\mathrm{JS}}$ instead. Furthermore, cluster-based divergences appear to be more stable, exhibiting smaller variances across random seeds. Collectively, our results suggest that cluster-based divergences may produce better metrics of text quality than string-based divergences. This motivates the question: Which components of natural language are captured by cluster-based distributions $p_c$, and which are overlooked by ignoring $p(\mathbf{w} \mid c)$ when computing the cluster-based divergences? Our next set of experiments aim to answer this.

## 6 PROBING CLUSTERS

To better understand the aspects of natural language that our cluster distributions encode, we must first understand how $\phi(\mathrm{PLM}(\cdot))$ partitions the string space $\mathcal{W}$. In other words, we must understand what components of natural language—e.g., semantics, syntactic attributes, or surface features—lead to strings being assigned to similar or different clusters. Such an analysis should provide a deeper insight into the actual similarity being measured by cluster-based divergences (while also revealing how such a metric might be gamed). To this end, we probe (Alain & Bengio, 2016) the clusters learned by the MAUVE algorithm for a number of linguistic attributes—including *subject matter*, *sentiment*, *prose style*, *word order*, *basic grammaticality* and *document length*—looking at how they affect both cluster assignment and the divergence scores between texts that differ in these characteristics. Notably, we probe cluster assignments directly—without relying on any diagnostic classifiers (Adi et al., 2017). Our probing analyses are thus exempt from most recent criticism against probing methodologies (Hewitt & Liang, 2019; Pimentel et al., 2020a;b; Ravichander et al., 2021; Elazar et al., 2021).

### 6.1 FINDING FEATURES $p_c$ ENCODES

**Setup.** We look at texts annotated with different attribute categories in order to explore correlations between the presence of these attributes and cluster membership. Specifically, we analyse texts': *sentiment*, *authorship*, and *topic* (using the Yelp Polarity, News Category, and 20 NewsGroup datasets, respectively). Further details on datasets are provided in App. D. For each of these classification datasets, we compute the cluster–category distributions that result from the MAUVE algorithm using the standard training split for the respective datasets; all evaluations are then performed on test splits. Explicitly, we first learn a partitioning $\phi(\cdot)$ of the embedding space (w.r.t a language model $\mathrm{PLM}(\cdot)$). Each cluster is then labelled with the *majority* category represented in that cluster by training examples; text categories in the test set are then predicted using this labelling, depending on which of the clusters the example falls into. For comparison's sake, we use four language models as $\mathrm{PLM}(\cdot)$: GPT-2 with `small`, `medium`, `large`, and `XL` architectures. Results using embeddings from BERT (Devlin et al., 2019) can be found in App. E. Further, we use two methods for learning clusters:

- $\phi(\cdot)$ **Learned on WebText.** Using the same procedure as in §5.2, we train PCA and $K$-means functions to partition the embedding space (again relying on WebText's test set for our data). This mimics the setting under which our partitions would be learned in practice.[9]

- $\phi(\cdot)$ **Learned on Training Set.** We instead train the PCA and $K$-means clustering functions on the analysed dataset's training set. This setting studies the partitioning that our clustering functions have the capacity to learn in an ideal setting, i.e., where the attribute in question is one of the main differentiating factors between texts.

---

[9]If no strings in the training set are assigned to a cluster, we label it with the overall majority category.

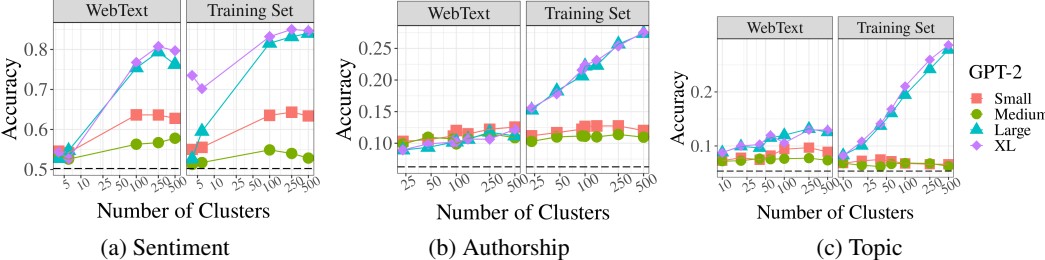

Figure 3: Accuracy when predicting different attributes of text from their cluster assignments. Assignments (i.e. $\phi(\cdot)$) are learned using text from either WebText, or the training set of the respective classification datasets. Dashed lines represent baseline accuracies, i.e., always guessing the majority class.

**Results.** In Fig. 3a, we see that, at least for large numbers of clusters, cluster assignment is indeed indicative of a text's sentiment. Interestingly, this is the case even when clusters are trained on data that is not particularly polar in sentiment (i.e., on WebText). On the other hand, we are only able to predict author and topic (with reasonable accuracy) when clusters are learned on text data with authorship and topic as distinguishing factors. These results indicate that, while writing style and subject matter are captured by the text embeddings, they likely were not being used as distinguishing features between corpora in our cluster-based divergences. We further see that, in all classification settings, the capacity to encode these analysed attributes appears to increase with model size, perhaps suggesting that the embedding spaces of larger models decompose along higher-level features of text.

## 6.2 How Text Features Impact $\Delta$

We next assess how changing different features of our evaluated text impacts divergence scores. Specifically, we look at the impact of: text truncation; article removal; stopword removal; sentence-level permutations; and word-level permutations.

**Setup.** We follow a similar setup to §5. In order to create a more controlled setting, though, we primarily consider human-generated text in these experiments (i.e., the $5k$ human-written articles in WebText's test set). We take the first 2500 articles of this dataset as our reference corpus $\left\{\mathbf{w}_n^{p_\mathbf{w}}\right\}_{n=1}^N$. We then use the remaining 2500 reference strings as the comparison corpus, i.e., in place of the model-generated text that we would typically evaluate $\left\{\mathbf{w}_n^{q_\mathbf{w}}\right\}_{n=1}^N$. In order to explore how changing specific features of text affects $\Delta$ w.r.t. the reference corpus, we then compute scores when making the following modifications to the comparison corpus:

- **No modification** ($p$). This is a baseline experiment where we keep the original strings unchanged.
- **Text Truncation** ($p_{\text{short}}$). We truncate texts to $1/3$ of their original length. This allows us to understand whether the divergence metrics pick up on differences in dataset length statistics.
- **Article Removal** ($p_{\text{no art}}$). We remove all articles ('a', 'an' and 'the') in the text. This allows us to understand whether the divergence metrics can distinguish between texts with or without basic levels of fluency and grammaticality.
- **Stopwords Removal** ($p_{\text{no stop}}$). We remove all stopwords (e.g., 'that' or 'so') in the text. This allows us to understand whether the divergence metrics can detect differing levels of syntactic coherence, rather than just focusing on content words.[10]
- **Sentence-level Permutation** ($p_{\text{swap}}$). We permute the first halves of texts (as delineated by sentences) *across* the entire corpus (i.e. randomly reassigning the strings' first halves). This allows us to understand whether the divergence metrics detect coherence.
- **Word-level Permutation** ($p_{\text{rand}}$). We randomly permute all words in a text. This allows us to understand whether the divergence metrics can only distinguish between bag-of-word level features.
- **GPT-2 Baseline** ($q$). As an extra baseline, we also use the first 2500 generations from GPT-2 XL on the WebText text-completion task, as in §5.

---

[10]Stopwords are defined as common words, such as "that" or "so", that primarily serve a syntactic function. We use the set of English stopwords defined by NLTK (Bird et al., 2009).

Figure 4: Divergence measures between two corpora: the reference text is unmodified while the comparison text undergoes perturbation. Higher values indicate a greater discrepancy according to $\Delta$.

**Results.** Fig. 4 shows that certain alterations to the evaluated text—such as completely removing articles—have almost no impact on its divergences from the reference corpora for various $\Delta$. In fact, text without any articles is judged as better than GPT-2 XL's by all of the cluster-based divergences (see Fig. 10 for a zoomed-in version). Further, while this perturbation undoubtedly affects the text's fluency, it has less of an effect on $\Delta$ than, e.g., truncating texts. This is arguably undesirable: A metric of text quality should place more emphasis on fluency than surface statistics, such as length.

On the other hand, our metrics deem text with stopwords removed as utterly different from the reference. Permuting words within texts has a similar effect, demonstrating that, at least to some extent, the embedding space captures notions of syntax and grammaticality, rather than pure unigram statistics. The increase in $\Delta$ shown when performing sentence-level permutations likewise suggests that the clusters delineate different levels of coherence to some extent. In Fig. 13 (in App. E), we perform an additional experiment where we again probe the clusters (as in §6.1), but for *surface* features of text this time, such as the percentage of stopwords and punctuation symbols in a text. There we see evidence that such features of text are not strongly encoded in the clustering scheme.

Perhaps surprisingly, when applied to cluster-based distributions, all of the studied $\Delta$ metrics rank the distance of the perturbed texts from the reference texts in the exact same order (this is clear in Fig. 10, a zoomed-in version of Fig. 4). For these perturbations, the $\Delta$ differ only in the magnitude of their outputs, further suggesting that the $\Delta_{\text{AUC}}$ metric itself is likely *not* critical for the effectiveness of MAUVE. One potential avenue for future research would be investigating whether different algorithms for discretisation of the embedding space create clusters that align with specific linguistic attributes; this could be a useful diagnostic for language generators' improvement.

This section's results—along with those of §6.1—suggest that divergences based on PLM embeddings are more sensitive to syntax- and coherence-related properties of the target text than to its superficial features. The opposite, however, might be said of our string-based distributions. These findings offer a potential explanation for the effectiveness of metrics that make use of PLM embeddings, such as MAUVE or BERTSCORE (Zhang et al., 2020). As current SOTA language generators already typically produce grammatical text, being invariant to surface statistics may perhaps be a feature—as opposed to a bug—when trying to assess the quality of the text they produce. Yet, this may also reveal potential ways in which such metrics can be gamed, bringing this paradigm's robustness into question.

# 7 CONCLUSION

In this paper, we analyse MAUVE, a recently-proposed automatic metric for language generator evaluation. While MAUVE correlates quite well with human quality judgements, it is unclear which of the metric's design choices are in fact responsible for its success—a shortcoming that impedes the further development of language generator evaluation metrics. We attempt to rectify this shortcoming. We first provide a general theoretical framework for the comparison of language generator evaluation metrics. Then through a series of empirical studies, we identify MAUVE's substitution of probability distributions over embedding-based clusters—in place of the traditional distributions over strings—as the attribute largely responsible for the metric's success. In order to better understand the nature of this improvement, we probe the clusters used by the density estimators, analysing what they ignore and what they emphasise about the input text. We find that, while distributions over clusters are sensitive to syntax- or coherence-level perturbations to the text, this is not the case for several surface-level perturbations. We thus conjecture that, by focusing on higher-level text features, cluster-based evaluation metrics may simply be better suited to rank high-performing models, and that this is a general paradigm worth further exploration.

## 8  ACKNOWLEDGEMENTS

We would like to thank the authors of MAUVE for sharing their human evaluation data with us. We would also like to thank Gábor Melis for his insightful feedback on an initial version of this paper, our anonymous reviewers for their help in improving our final draft, and Luca Malagutti for detailed feedback on clarity and presentation.

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

## A   A MONTE CARLO ESTIMATOR FOR BACKWARD, JS, AND AUC DIVERGENCES

Consider a Monte Carlo estimator for $\Delta_{\leftarrow}$:

$$\Delta_{\leftarrow}(p_{\mathbf{w}}, q_{\mathbf{w}}) \approx \widehat{\mathrm{KL}}(q_{\mathbf{w}} \,||\, p_{\mathbf{w}}) = \frac{1}{N} \sum_{n=1}^{N} \log \frac{q_{\mathbf{w}}(\mathbf{w}_n^{q_{\mathbf{w}}})}{p_{\mathbf{w}}(\mathbf{w}_n^{q_{\mathbf{w}}})} \tag{14}$$

where $\mathbf{w}_n^{q_{\mathbf{w}}} \sim q_{\mathbf{w}}$. We can easily sample from $q_{\mathbf{w}}$. However, computing Eq. (14) requires knowledge of $\log p_{\mathbf{w}}$, which we do not have. Thus we resort to other techniques for estimating these divergences, as discussed in §3.

## B   STRING VS. CLUSTER-BASED KULLBACK–LEIBLER DECOMPOSITION

The decomposition in Eq. (13) can be shown as follows:

$$\mathrm{KL}(p_{\mathbf{w}} \,||\, q_{\mathbf{w}}) = \sum_{\mathbf{w} \in \mathcal{W}} p(\mathbf{w}) \log \frac{p(\mathbf{w})}{q(\mathbf{w})} \tag{15}$$

$$\overset{(1)}{=} \sum_{c=1}^{K} \sum_{\mathbf{w} \in \mathcal{W}} p(c, \mathbf{w}) \log \frac{p(c)\, p(\mathbf{w} \mid c)}{q(c)\, q(\mathbf{w} \mid c)}$$

$$= \sum_{c=1}^{K} \sum_{\mathbf{w} \in \mathcal{W}} p(c, \mathbf{w}) \left( \log \frac{p(c)}{q(c)} + \log \frac{p(\mathbf{w} \mid c)}{q(\mathbf{w} \mid c)} \right)$$

$$= \underbrace{\sum_{c=1}^{K} p(c) \log \frac{p(c)}{q(c)}}_{\text{Marginalise over all } \mathbf{w}} + \sum_{c=1}^{K} \sum_{\mathbf{w} \in \mathcal{W}} \underbrace{p(\mathbf{w} \mid c) p(c)}_{= p(c, \mathbf{w})} \log \frac{p(\mathbf{w} \mid c)}{q(\mathbf{w} \mid c)}$$

$$= \sum_{c=1}^{K} p(c) \log \frac{p(c)}{q(c)} + \mathbb{E}_{p_c} \left[ \sum_{\mathbf{w} \in \mathcal{W}} p(\mathbf{w} \mid c) \log \frac{p(\mathbf{w} \mid c)}{q(\mathbf{w} \mid c)} \right]$$

$$= \mathrm{KL}(p(c) \,||\, q(c)) + \underbrace{\mathrm{KL}(p(\mathbf{w} \mid c) \,||\, q(\mathbf{w} \mid c))}_{\geq 0}$$

$$\geq \mathrm{KL}(p_c \,||\, q_c)$$

where again (1) follows from the fact that $p(c, \mathbf{w}) = p(\mathbf{w})$, which is true because the cluster assignment is deterministic, i.e.:

$$p(c \mid \mathbf{w}) = \mathbb{1}\left\{ c = \phi(\mathsf{PLM}(\mathbf{w})) \right\} \tag{16}$$

Thus, $\mathrm{KL}(p(c) \,||\, q(c))$ provides a biased estimate of $\mathrm{KL}(p_{\mathbf{w}} \,||\, q_{\mathbf{w}})$.

## C   RELATED WORK

Over the years, a number of evaluation metrics have been proposed for language generation tasks (such as translation and summarization); the most well-established and commonly-used include BLEU (Papineni et al., 2002), ROUGE (Lin, 2004) and METEOR (Banerjee & Lavie, 2005). However, these metrics—which rely on $n$-gram statistics—have been shown to correlate poorly with human judgement (Reiter, 2018). Recent metrics have improved upon these correlations with more advanced techniques, e.g., BEER (Stanojević & Sima'an, 2014), MoverScore (Zhao et al., 2019) and BLEURT (Sellam et al., 2020). These metrics, though, are intended for language generation tasks with a strict set of reference texts. While reasonably effective for directed generation tasks, they do not transfer well to the open-ended domain.

For tasks in which there is not a clear reference, e.g., story generation, basic statistics are typically employed to provide a preliminary evaluation of generated text. Such statistics include $n$-gram

repetitions (Welleck et al., 2020), Zipfian coefficient (Holtzman et al., 2020), or the perplexity of generated text (Fan et al., 2018). Final assessments of language generation systems are still often performed using human evaluations, as automatic metrics on their own have not proven sufficient for differentiation between top-end language generation systems.

Automatic evaluation metrics for language models based on statistical divergences have been proposed by a number of different authors. For instance, Meister & Cotterell (2021) assessed the quality of language models while using a number of divergences between distributions over surface statistics in text corpora. Xiang et al. (2021) propose the approximation of distributions over strings with distributions in the embedding space in standard divergence metrics. Pillutla et al. (2021) present MAUVE—the object of study of this work—which is based on a new divergence metric inspired by the information frontier divergences Djolonga et al. (2020).[11] Besides proposing this AUC divergence metric, Pillutla et al. (2021) also propose a new way to approximate it using clusters over word embeddings. We provide an analysis of this paradigm, showing that in practice, we should expect this method to provide a poor estimation of the intended quantity. We go on to perform empirical experiments to identify the proposed use of distributions over clusters itself is likely responsible for the metric's success, rather than characteristics of the new divergence. We see this work as complementary to Pillutla et al. (2021), providing deeper and more comprehensive insights into the metric's inner workings.

## D  EXPERIMENTAL SETUP

**Embedding Models.** To obtain text embeddings for our input strings, we use several different PLM. Namely, we use the four sizes of GPT-2 (Radford et al., 2019), as well as BERT-base and BERT-large (both cased; Devlin et al., 2019). For the former, we use the embedding of the final token. For the latter, we use the embedding associated with the special `CLS` token. In both cases, we additionally present results using the average across token embeddings in App. E. All texts are truncated to 512 tokens (for experiments in §6.2, this truncation is performed before any manipulations of the text), in order to ensure that all models can process the input text in their context.

For our probing analysis in §6, we employ the following datasets:

- **Sentiment.** To analyse sentiment, we use Yelp Polarity (Zhang et al., 2015), a dataset extracted from the Yelp Dataset Challenge 2015, which contains binary sentiment classifications for highly polar Yelp reviews. We use $10k$ examples randomly sampled from the training set and $5k$ examples randomly sampled from the test set.

- **Authorship.** To analyse authorship, we use News Category (Misra & Grover, 2021; Misra, 2022), a dataset consisting of $200k$ news headlines from the years 2012–2018 obtained from HuffPost. We scrape entire articles from the URLs provided by the dataset. We only use the subset of articles for which the article's author has $\geq 400$ articles within the dataset, giving us a training set of $32k$ and a test set of $6k$ with 46 unique authors.

- **Topic.** To analyse topic, we use the 20 NewsGroup dataset, which contains $18k$ newsgroups posts (partitioned into train and test sets using the original splits) on 20 topics, such as subcategories of science, politics and religion. The distribution over text topics is relatively uniform.

---

[11]Specifically, Djolonga et al. proposed a framework that measures the trade-off between precision and recall using Rényi divergences.

# E    ADDITIONAL EXPERIMENTS

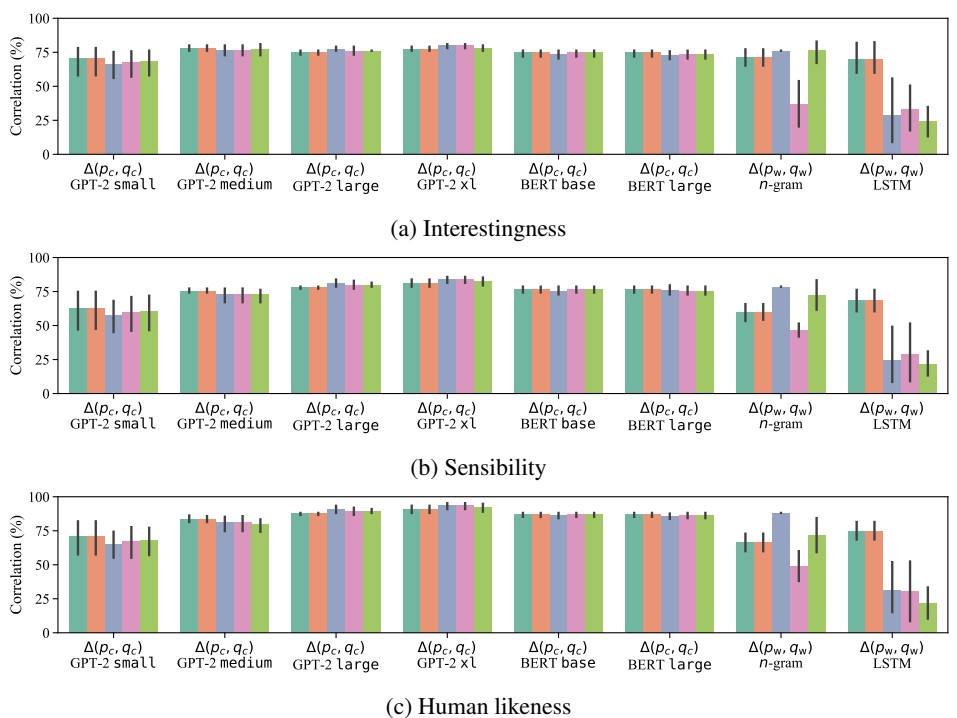

(a) Interestingness

(b) Sensibility

(c) Human likeness

Figure 5: Correlations between string- and cluster-based divergences and human judgement scores using a number of estimators (with different PLM($\cdot$) when defining $p_c$, or language models for $p_{\mathrm{w}}$). Legend: $\triangle_{\exp}$ **in dark green**; $\triangle_{\rightarrow}$ **in orange**; $\triangle_{\leftarrow}$ **in blue**; $\triangle_{\mathrm{JS}}$ **in pink**; $\triangle_{\mathrm{AUC}}$ **in lime green**.

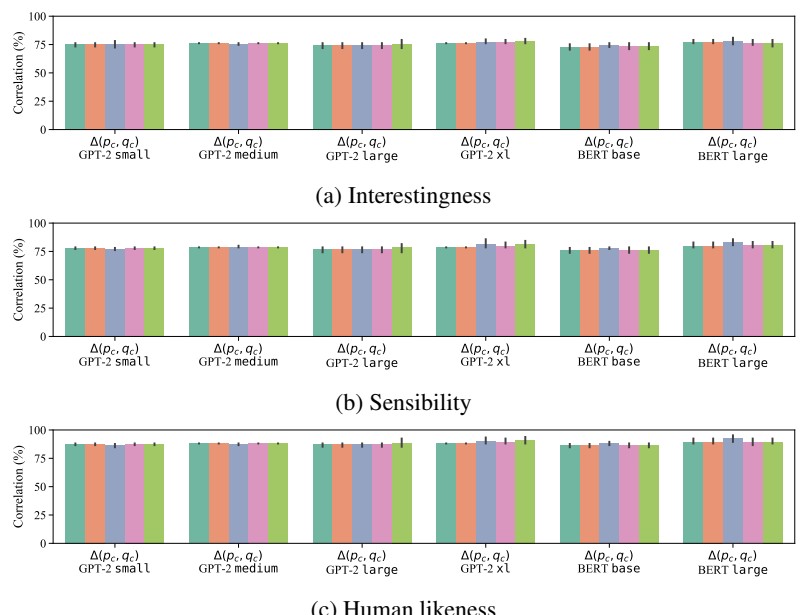

(a) Interestingness

(b) Sensibility

(c) Human likeness

Figure 6: Correlations between cluster-based divergences and human judgement scores using a number of PLM($\cdot$) to define $p_c$. In this figure, we use the average embedding per sentence produced by a PLM($\cdot$) to compute $p_c$, as opposed to the final embedding. Legend: $\triangle_{\exp}$ **in dark green**; $\triangle_{\rightarrow}$ **in orange**; $\triangle_{\leftarrow}$ **in blue**; $\triangle_{\mathrm{JS}}$ **in pink**; $\triangle_{\mathrm{AUC}}$ **in lime green**.

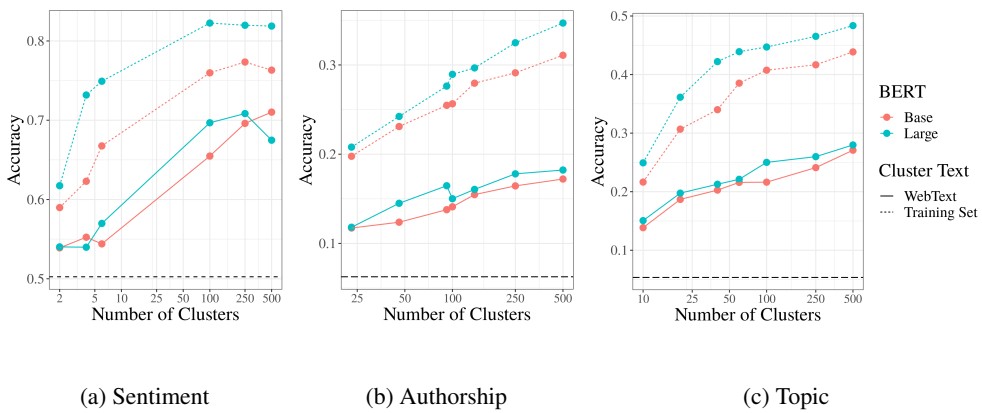

(a) Sentiment        (b) Authorship        (c) Topic

Figure 7: Accuracy when predicting different attributes of text from their cluster assignments. Same plot as Fig. 3 albeit using embeddings from BERT.

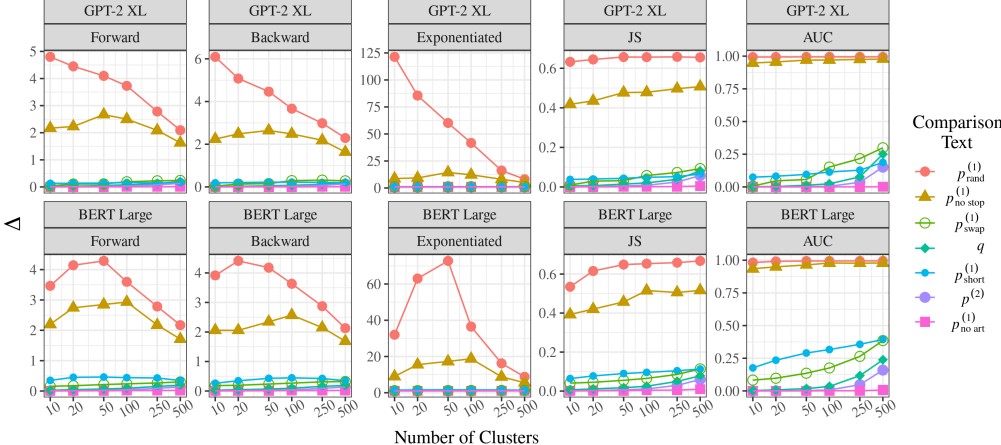

Figure 8: $\Delta_{\mathrm{AUC}}$ scores between reference text and alternate text distributions as a function of number of clusters used to estimate $\widehat{p}_c$ and $\widehat{q}_c$. Importantly, in this figure, we compare the first 2500 sentences ($p^{(1)}$) in the human-generated WebText test set to these same strings, but under the proposed interventions. I.e., for all points corresponding to an (altered) distribution $p^{(1)}$, we estimate our cluster distributions on the same set of human-generated sentences, but where the strings in one group have been intervened on. The baseline distribution $p^{(2)}$ still represents the final 2500 sentences in WebText (as in the original plot); and baseline distribution $q$ is text sampled from GPT-2 XL.

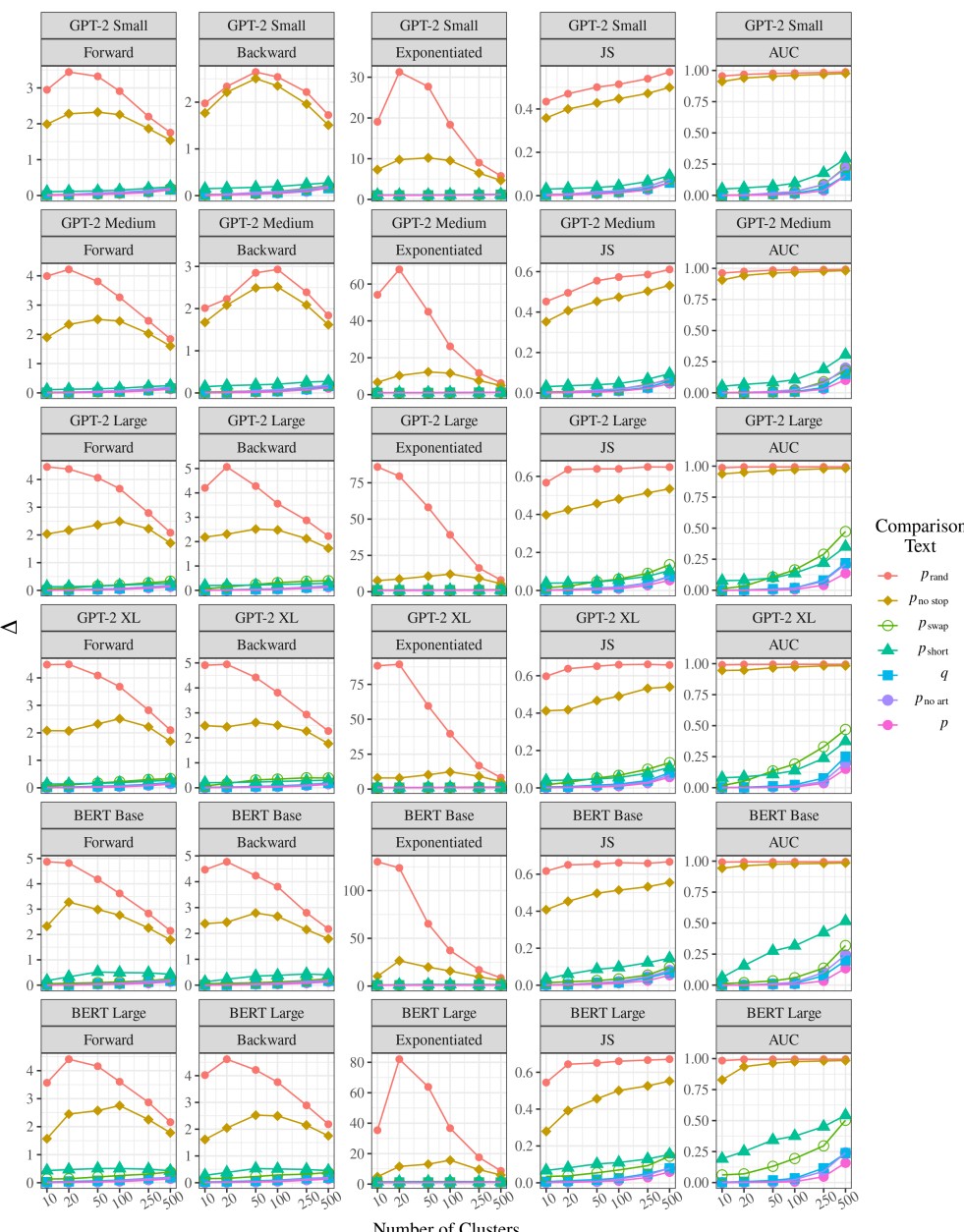

Figure 9: $\Delta$ scores between reference text and alternate text distributions as a function of number of clusters used to estimate $\widehat{p}_c$ and $\widehat{q}_c$. Results shown for embeddings produced using multiple LMs

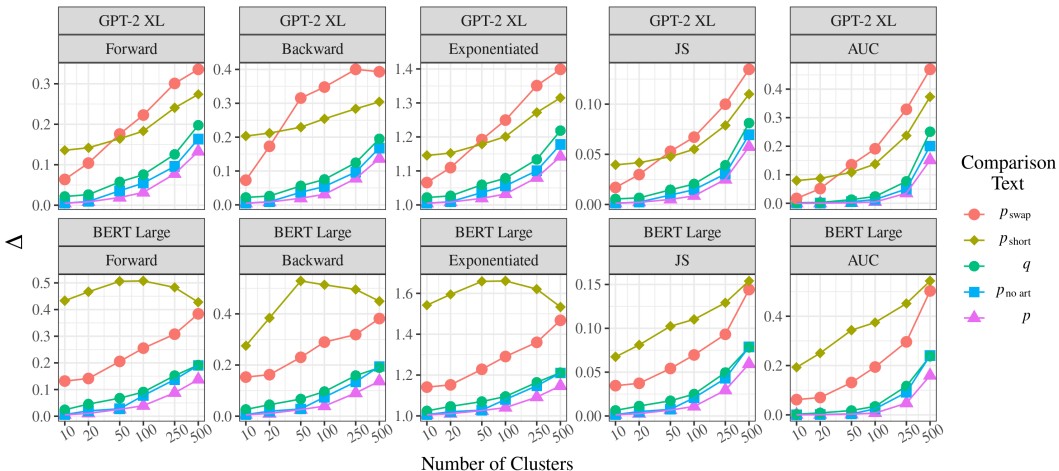

Figure 10: Zoomed in version of Fig. 4 to give a closer look at different scores assigned to texts manipulated in different ways.

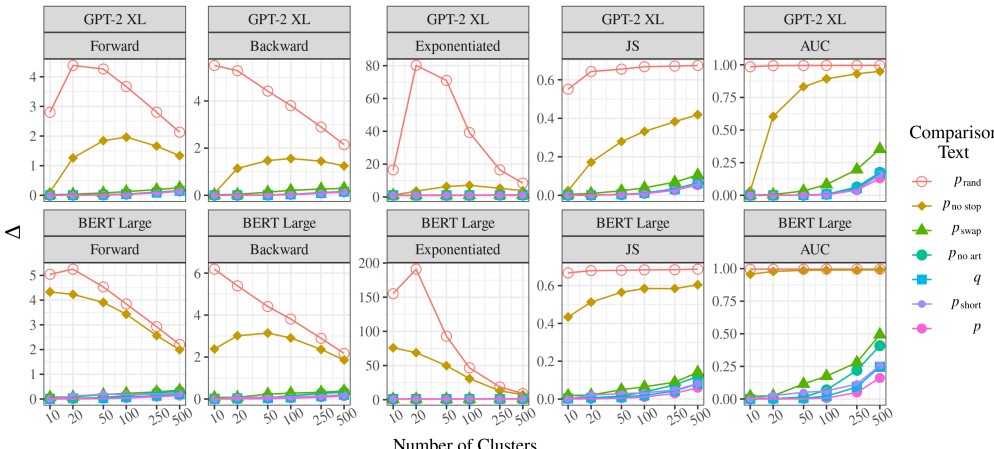

Figure 11: Version of Fig. 9 that uses the *mean* of the contextual embeddings from a text to form clusters.

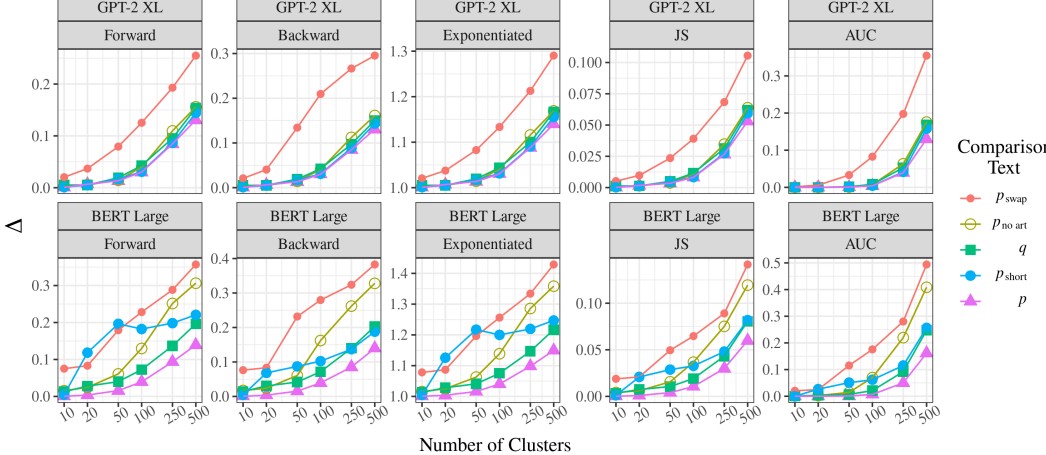

Figure 12: Version of Fig. 10 that uses the *mean* of the contextual embeddings from a text to form clusters.

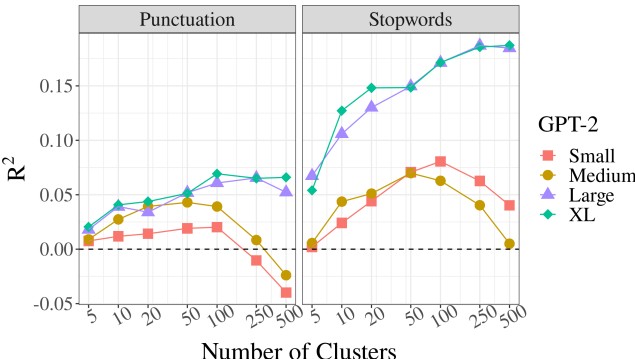

Figure 13: $R^2$ when using cluster assignments to predict % of tokens in a text that are either punctuation or stopwords. Setup follows that of §6.1, albeit using solely the WebText dataset to train our clustering functions in this setting. We compute the average percentage of stopwords or punctuation per cluster in half of our strings and use these pre-computed averages when predicting the percentages in the other half, computing this prediction's $R^2$ (i.e. the percentage of explained variance). We see that larger PLMs—which are often claimed to provide better representations of language—do encode more information about such surface features than smaller models. This could simply be due to the fact that the embeddings from larger PLMs are typically of a larger dimension and, thus, have the capacity to encode additional (perhaps "less critical") attributes of text. While these attributes do not appear to be differentiating factors when partitioning the embedding space into a small number of clusters, they become relevant when partitioning into a larger number of clusters. Even with several clusters and large PLMs, though, the $R^2$ values we find are still quite small, at around 0.20.

