# OpenReview forum: "On the Usefulness of Embeddings, Clusters and Strings for Text Generation Evaluation"
_ICLR.cc/2023/Conference — ICLR 2023 notable top 25%_

### Official Review · Reviewer_H8v4 · 2022-10-24

**Confidence:** 3
**Correctness:** 3
**Technical Novelty And Significance:** 4
**Empirical Novelty And Significance:** 4
**Recommendation:** 6

**Clarity, Quality, Novelty And Reproducibility:**

This work is about to analyze one successful prior metric's performance and to investigate what is the main reason of its success. This kind of papers which pay attention mostly to the analysis and conducts different experiments to probe the problem in details can be useful and indeed have been less paid by the researchers in evaluation field.

**Strength And Weaknesses:**

The comprehensive analysis of different divergences and estimations of probabilities for various models with different sizes is the major strength of the paper that gives value to such type of analysis-based papers. The order of discussed contents in the paper looks very neat and clear to me specifically for the first three sections.

There are some minor issues in the experimental section, that are summarized here:

In string-based approximations, only ngram-based model has been considered, while to have a complete analysis it would be good to consider other type of generative models such as BART even though its decoder is GPT but still can be informative and good for comparison.

In section 5.1, authors have shown that cluster-based approximations are not accurate estimators for original distribution, thus the question that arises here is that why in section 5.2 it is shown that even though they are not accurate using them have good impact on different divergences and consequently the metric's performance. Some further explanations can be beneficial.

In section 5.2, different string-based and cluster-based approximations have been compared, the Monte Carlo estimations for specific divergences (forward, exp)  can also be useful to be involved.

One interesting experiment that can be added to the paper is the comparison between the inductive bias that can be added by different approximations for divergences to show the pros and cons of each approximations for the divergences.

In cluster-based approximations only last word embeddings are leveraged, is there any reasons behind it.

I strongly suggest to use shapes for plots to be readable in any modes.

Based on results in Figure 2, is there any probing of why substituting string-based approximations with cluster-based approximations in AUC divergence results in lower correlation enhancement specifically in interestingness evaluation of texts.

Two motivation questions in section 5 is not very intuitive. Some explanation can better lead the reader to the next experiments.

**Summary Of The Paper:**

The paper is an analysis of Maude metric which computes the closeness of distribution of human-written text with the distribution of texts generated by generative models using the AUC divergence. In this paper, it is shown that the success of Maude is not because of the AUC divergence rather it is due to the cluster-based approximations of the original distributions using pre-trained LM embeddings.

Authors show that different divergences based on cluster-based approximations have positive impact on the metric's performance. They do a comprehensive probing on these cluster-based approximations and find out that many syntactic and coherence level features such as sentences-substitutions have higher impact on the cluster assignment of strings rather than surface-level features. These can be good features for evaluating SOTA generative models that mostly generate grammatically correct text while texts are not always syntactically correct and coherent.

**Summary Of The Review:**

In overall the motivation, research problem and conducted analysis are written quite fluently, however there are some not clear parts that have been mentioned in the weakness section.

---

> ### Author Response · Authors · 2022-11-10
> **Response**
>
> Thank you for your feedback. We address each of the reviewer points below.
> * In string-based approximations, only ngram-based model has been considered.
> We would like to clarify a possible misunderstanding here: We also experiment with LSTMs for our string-based approximations of the distribution; results can be seen in Figure 5 (in Appendix E). While we agree that it would be interesting to try other models here, we are limited by the amount of data in the evaluation set that can be used to estimate the model (as discussed in section 3.2).
>
> * In section 5.1, authors have shown that cluster-based approximations are not accurate estimators for original distribution, thus the question that arises here is that why in section 5.2 [... they] have good impact on the metric's performance.
> We likewise find the results of 5.2 confusing, given that we see in 5.1 that cluster-based approximations serve as poor estimates of string distributions. Section 6 aims to answer why this is the case via probing, essentially trying to uncover what properties of language cluster-based divergences are encoding. Our results lead us to conclude that the fact that pretrained language model embeddings mostly encode higher-level features of text (in opposition to surface-level features) is helpful.
>
> * In section 5.2, different string-based and cluster-based approximations have been compared, the Monte Carlo estimations for specific divergences (forward, exp) can also be useful to be involved.
> We note that using these estimators with the original language generator distributions q will lead to infinite divergences for all models using nucleus sampling. This led us to choose not to run this analysis. We can try to compute and send these results here by the end of the discussion period, though, if the reviewer wants to see them. We also note that the cluster-based divergences can actually be computed exactly, without monte carlo estimation. Computing them with or without monte carlo estimation results in divergence scores with very similar correlations with the human judgements.
>
> * One interesting experiment that can be added to the paper is the comparison between the inductive bias that can be added by different approximations for divergences to show the pros and cons of each approximations for the divergences.
> This sounds like an interesting idea, but we are not sure how to analyze that beyond what we tried in the paper. We believe the Monte Carlo estimators, for instance, should not strongly bias the results towards any specific direction, just make them more noisy. The plug-in estimators should lead to different biases depending on the used approximator family π (we have experiments with an n-gram family in the main text, and LSTMs in the appendix). The cluster-based estimators biases will also likely depend on the choice of PLM, for GPT-2 (and BERT in the supplemental materials) we tried to shed some light into these biases through our probing experiments.
>
> * In cluster-based approximations only last word embeddings are leveraged, is there any reasons behind it.
> We use only the last embedding from GPT-2 as this was the method employed by Pillutla et. al (for BERT, we used the embedding associated with the CLS token). We found that using the average of all embeddings did not have a large effect on results; we will add these results to the appendix in the coming days.
>
> * I strongly suggest to use shapes for plots to be readable in any modes.
> We will add new plots with shapes in the coming days as well.
>
> * Based on results in Figure 2, is there any probing of why substituting string-based approximations with cluster-based approximations in AUC divergence results in lower correlation enhancement specifically in interestingness evaluation of texts.
> Unfortunately, we could not come up with such an experiment. Another question we find interesting is why the backward divergence seems to work best with the ngram-based distributions.
>
> * Two motivation questions in section 5 is not very intuitive.
> We tried to clarify these in the new version (just uploaded) of our paper.

---

> > ### Comment · Reviewer_H8v4 · 2022-11-14
> > **Feedback**
> >
> > Thanks authors for the response and adding the details to make the paper be more clear. Even though there are still some open questions here but I do think they can be left for the further research. In overall, I am satisfied with this work.

---

### Official Review · Reviewer_mdcX · 2022-10-24

**Confidence:** 4
**Correctness:** 3
**Technical Novelty And Significance:** 4
**Empirical Novelty And Significance:** 4
**Recommendation:** 8

**Clarity, Quality, Novelty And Reproducibility:**

The paper is novel but the clarity needs further improvements. For example:
1. in section 5.2, it is better to briefly introduce what the generation task is and how it was evaluated.
2. in the probing experiments, it is better to clearly indicate which modification is aiming at probing which linguistic information.

**Strength And Weaknesses:**

Strength:
1. This paper revisits one of the STOA NLG evaluators either theoretically or empirically. This explains which component of MAUVE really works and helps the future development of NLG evaluation research.
2. Probing experiment helps readers understand MAUVE better.

Weaknesses:
I found no major risk in accepting the present paper. I have only one minor concern: I somewhat feel the probing results are not in line with the correlation analysis. Concretely, from figure 4, it appears that different divergence measures are sensitive to different linguistic information. For example, Exponentialed is sensitive to p_{rand} while AUC is sensitive to p_{swap}. However, they all have similar correlations with human scores. My understanding is that this is might be a result of the limitation of the human scores you used. More specifically, the probing experiments target syntax and coherency related information, while human scores in Figures 4 (i.e., interestingness, sensibility, and humanlikeness) are all from pragmatic aspects. More explanation is welcome here.

**Summary Of The Paper:**

The current paper focuses on revisiting one of the state-of-the-art NLG evaluation metrics, namely MAUVE. This paper proves that the good performance of MAUVE is coming from the cluster-based approximation it used but not the AUC divergence. Additionally, this paper conducts a probing experiment to understand what linguistic information has each cluster-based evaluator learnt.

**Summary Of The Review:**

The paper is generally is in a very good shape and definitely should be accepted, but there are still some minor issues that are yet to be clarified or explained.

---

> ### Author Response · Authors · 2022-11-10
> **Response**
>
> Thank you for the recommendations to improve the clarity of our paper. We have now implemented them! Specifically, we tried to clarify the motivation behind each perturbed distribution in section 6.2, and to clarify how text was generated from our language generators in section 5.
>
> Further, we added a zoomed in version of our Fig. 4 to the appendix (it is Figure 9 in the updated version). In this figure it becomes clearer that all of the divergence metrics actually rank the perturbed distributions in the same order. We agree that investigating whether any of these divergences should be particularly sensitive to specific linguistic attributes would be interesting, though. Further, the impact of how human evaluations are executed on these rankings is also deeply interesting, although we did not have the insight to come up with an experiment to test that question exactly yet. We added this discussion as an extra paragraph to our Section 6.2.

---

> > ### Comment · Reviewer_mdcX · 2022-11-10
> > **human evaluation**
> >
> > Thanks for your response. It answers all my concerns.
> >
> > Regarding human evaluation, maybe something that is worth reading:
> > 1. Twenty Years of Confusion in Human Evaluation: NLG Needs Evaluation Sheets and Standardised Definitions
> > 2. Human evaluation of automatically generated text: Current trends and best practice guidelines

---

### Official Review · Reviewer_X7E3 · 2022-10-25

**Confidence:** 4
**Correctness:** 4
**Technical Novelty And Significance:** 4
**Empirical Novelty And Significance:** 4
**Recommendation:** 8

**Clarity, Quality, Novelty And Reproducibility:**

See strength and weaknesses.


**Strength And Weaknesses:**

Strengths:

1. Very well organized and clearly presented.

2. The step-by-step rationale for the MAUVE approach (and variants), although not novel, is extremely thorough and provides useful structure for organizing work in this area.

3. The finding about the central utility of LM-derived clusters is very significant, given the prominence of the MAUVE paper. This is something that needs to be publicized, and that will stimulate further research in a crucial area.

4. The experiments into the nature of the clusters yield interesting insights, particularly that metrics like MAUVE are relatively insensitive to surface features, and might be vulnerable to gaming as a result.

Weaknesses:

The evaluation is performed only with variants of GPT-2. It would be interesting to see whether the results carry over to more recent models such as GPT-3 or PaLM.


**Summary Of The Paper:**

This paper analyzes the performance of the recently-introduced and influential MAUVE metric for evaluating the quality of automatically-generated text by comparing it to human-generated text at the distribution level. It begins by setting out the rationale for the MAUVE approach, starting with difficulties in applying standard divergence measures to this particular evaluation problem and motivating the use of surrogate distributions such as MAUVE’s embedding-cluster-based multinomials. It then demonstrates that these surrogates provide a biased approximation to the true KL divergence (an ingredient in the AUC divergence used by MAUVE).

Experiments show that cluster-based surrogates nevertheless have much better correlation with human judgments than do ngram LMs. This finding is independent of the particular divergence metric used, implying that the use of clusters rather than AUC divergence is the key factor in MAUVE’s performance. Further experiments probe the nature of the clusters, finding that they correspond to sentiment, but only weakly to style and topic. Finally, when comparing perturbed and natural human samples using the cluster approach, surface features such as articles and punctuation are found to be largely irrelevant.

**Summary Of The Review:**

The paper is technically very strong, and it presents important experimental results in the key area of quality evaluation for text generation by LLMs. The insights into how MAUVE works need to be publicized.

---

> ### Author Response · Authors · 2022-11-10
> **Response**
>
> Thank you for your feedback! We agree that performing evaluations with additional language models would strengthen the empirical results of the paper. We do provide results using variations of BERT (both base and large) as our pretrained language models in the Appendix (in section E); these additional results include both correlations with human judgements when using BERT embeddings for the clustering scheme, as well as a probing analysis on it. If we understood this comment correctly, we also agree that running our evaluations on text generated by GPT-3 and PaLM would be interesting; unfortunately, the need to run new human evaluations constrains our ability to do this within the discussion period.

---

> > ### Comment · Reviewer_X7E3 · 2022-11-22
> > **Acknowledging response**
> >
> > That makes sense; thanks for your response.

---

### Official Review · Reviewer_X1r9 · 2022-10-26

**Confidence:** 4
**Correctness:** 4
**Technical Novelty And Significance:** 3
**Empirical Novelty And Significance:** 3
**Recommendation:** 6

**Clarity, Quality, Novelty And Reproducibility:**

The paper is well written, the proofs and formulations are quite correct to my best judgment.

Documentations of experimental work in section 5 and 6 could have used more clarity and being specific about experiments details, not only this can help in reproducibility but also help judging the correctness of the results in the experiments.



**Strength And Weaknesses:**

Strengths:
- The paper is well written the related work is quite educational, instead of listing papers authors make the effort of laying a theoretical ground for comparing difference divergence metrics and ways to approximate their intractability.
- The paper shows theoretical and empirical drawbacks of the mauve score, and show why it does work in practice
- I find the findings from section 5.1 and 5.2 to be interesting, mainly that methods that correlate with the real probability distributions $\hat{p}_w$ and $\hat{q}_w$ are not necessarily what is best to correlate with Human judgements $\hat{p}_c, \$hat{q}_c. This opens more research questions and could have made this paper have a wider scope.

Weaknesses:

The scope of the paper is quite narrow, and solely serves as a critique to the Mauve score paper which was published last year and yet not so widely adopted in LM evaluation. It is not clear how those conclusions could inspire future works for language model evaluation.




**Summary Of The Paper:**

The paper is about evaluation of open-ended generation of LMs, in particular the Mauve (pillutla et al.) metric.

Evaluation of open ended text generation takes a distributional format. It is framed in this paper Divergence(p_w, q_w). This task has two sub-tasks, (1) Density estimation of p_w and q_w (when necessary). (2) calculating a distribution divergence metric of choice and -if necessary- approximate p_w or q_w to calculate this divergence metric tractably.


The paper starts with laying a common formalism to compare these divergence metrics and notably revisits the AUC divergence metric used in the Mauve metric. This metric is intractable to calculate in practice and therefore P_true and P_model are replaced by multinomial probability over clusters of sentence embeddings using an external encoder.

This paper is a critique of the Mauve metric both theoretically and empirically, mostly the replacement of p_w and q_w with its clustering based approximation:
- Section 4: it shows that the approximation kl(p_w,q_w) to kl(p_c,q_c) is biased
- Section 5.1: it shows empirically that string based density estimations (using basic n-gram lm) of p_w and q_w correlations better with original data distribution than their cluster based counter parts
- Section 5.2: Despite the results shown in section 5.1, authors show that divergence metrics correlate *correlate better with human judgements* when cluster based density estimation is used as an approximation.
- Section 6: by probing the clusters authors show that clustering based on sentence embeddings (the core component of the clustering based density estimation approx of mauve) are more oriented to capture global features such as sentiment, authorship and topic and less sensitive to surface modification which is probably why they correlates with human judgements overall since most recent language models don’t suffer from disfluency,  although this could be a way to game the mauve score


**Summary Of The Review:**

The paper overall serves as a good critique of the mauve score, empirical and theoretical findings in the paper seems to be solid, however the overall scope of the paper is quite narrow and focuses only on critiquing one paper.

---

> ### Author Response · Authors · 2022-11-10
> **Response**
>
> We thank the reviewer for their thoughtful feedback. As we understand this review, the main weakness in our paper would be its scope, which the reviewer evaluates to be narrow. While we partly agree with this statement—as our paper does focus on critically evaluating Mauve—we do not believe this makes the work overly narrow for the following reasons:
> * Beyond just a critique of Mauve, we believe this work should also serve as a general framework for the creation and analysis of future text generator evaluation metrics. These metrics currently lack a unifying framework, which makes their theoretical comparison difficult, i.e., it is difficult to understand the commonalities and differences between successful and unsuccessful metrics without such a framework. Reviewer X7E3, for instance, has commented that this framework “provides useful structure for organizing work in this area” and you also note that we “make the effort of laying a theoretical ground for comparing difference divergence metrics and ways to approximate their intractability,” which we believe supports this opinion.
> * Second, Mauve is arguably reasonably popular; it already has 36 citations, within less than a year of being published in NeurIPS. Further, it has seen a large increase in popularity in recent months. For example, it is becoming widely adopted amongst papers in ACL’s language generation track, a trend which will likely continue given the boost in popularity that often follows outstanding paper awards. Finally, it was recently made available in the HuggingFace library (https://huggingface.co/spaces/evaluate-metric/mauve), which is likely to further increase its adoption.
> * Lastly, while we focus on Mauve as an evaluation metric in our analysis, our results should have implications to other language generation evaluation metrics. BERTScore, for instance, is similarly based on evaluating the similarity between the contextualised embeddings of generated and reference sentences. Our probing analysis, which highlights both positive and negative features of using GPT’s embeddings for language generation evaluation (as well as BERT’s, in the Supplemental Materials) should similarly have implications for why that metric works.
>
> We added the first and last points above to our manuscript with the hopes that this clarifies the broader implications.
>
> We have also clarified our experimental setup in sections 5 and 6—or at least we believe it should be clearer now. Thank you for pointing out this oversight. Please let us know if you have suggestions of how it could be further improved!

---

### Author Response · Authors · 2022-11-10
**Revisions in response to reviewer feedback**

The main changes we implemented in the manuscript were:
* We added a few extra sentence (to the intro and conclusion) discussing the positive implications of providing a general framework for comparing text generator evaluation metrics.
* We changed the description of our experimental setups (in both sections 5 and 6) to try and clarify them. We are happy to work further on any points that the reviewers feel could be improved.
* We added a zoomed in version of Figure 4 to the appendix (as Figure 9) which reveals the similarities in the trends shown by each of  the divergence metrics. We added a paragraph commenting on these results in section 6.2.
* We also moved our old Figure 5 (on which we probed the clusters for surface-level statistics, such as the percentage of stopwords or punctuations in a sentence) to the appendix to get space to effect the changes above.
* We added a new version of some of our plots with shapes/styles added in.


The changes we still plan to execute, but haven't yet:
* Results using the average over GPT's word embeddings, instead of the embedding of the last word per sentence.

---

> ### Author Response · Authors · 2022-11-18
> **Extra revision in response to reviewer feedback**
>
> We have just added to our manuscript new results using the average over GPT's word embeddings, instead of the embedding of the last word per sentence. We again thank the reviewers for their detailed feedback.

---

### Decision · Program_Chairs · 2023-01-20

**Decision:**

Accept: notable-top-25%

**Justification For Why Not Higher Score:**

The paper provides insights into this metric and will motivate further work in this challenging area. The experimentation could be extended in the future to include other, more recent language models as well.

**Justification For Why Not Lower Score:**

All reviewers and I agree on the usefulness of the insights the paper proposes on an important and challenging area.

**Metareview: Summary, Strengths And Weaknesses:**

Mauve is a recently proposed metric for natural language generator evaluation, and it works by directly comparing the learnt distribution from a text generation model to the distribution of human-written text, using divergence. This paper analyzes the performance of this metric and provides insights into how this metric works by using surrogate distributions, such as embedding-cluster-based multinomials, due to the difficulty of standard divergence measures for this evaluation problem. The paper shows that Mauve works well not because of the area under the curve divergence, but mainly due to these cluster-based approximations of the original distributions using embeddings from pre-trained language models. The paper is easy to read, it clearly motivates the analysis and includes experimentation that provides insights into the nature of clusters.

**Note From Pc:**

if the above contains the word "oral" or "spotlight" please see: "oral" presentation means -> notable-top-5% and "spotlight" means -> notable-top-25%. As stated in our emails, we are disassociating presentation type from AC recommendations